# Examining the Role of Leisure in Navigating Spousal Death: A Phenomenological Multi-Case Study of Widowhood

**Thomas P. Sweeney and Jennifer Zorotovich \***

School of Human Ecology, Georgia Southern University, Statesboro, GA 30458, USA;
tsweeney@georgiasouthern.edu
\* Correspondence: jzorotovich@georgiasouthern.edu

**Abstract:** The current study will explore the post-pandemic bereavement window of widowed caregivers who experienced higher degrees of social isolation and the ways in which leisure was used to cope. A phenomenological multi-case study technique was used to explore the "multiple realities" among female caregivers whose social convoys suffered pre-pandemic because of spousal caregiving and were further impacted by COVID-19 mitigation strategies during the pandemic. Initial analyses utilized the two-cycle approach, as defined by Saldana. In the initial cycle, the researchers independently immersed themselves in the raw data gathered during participant interviews and engaged in open coding to discern concepts, patterns, themes, and categories associated with the multiple realities among the widows whose leisure participation was affected due to the strain of spousal caregiving and exacerbated by COVID-19 mitigation strategies. After completing the first cycle, the researchers then moved to the second phase, thematic analysis, which sought to develop a more structured framework by which to understand the data by identifying, analyzing, and reporting patterns (e.g., themes) within it. A qualitative comparison technique was then employed to deepen our understanding of individuals' lived experiences. Thematic findings revealed three areas of similarities: Social isolation from caregiving, using leisure to self-focus and explore future possibilities, and core groups.

**Keywords:** recreation; leisure; widowhood; coping; COVID-19

## 1. Introduction

The tangible benefits of leisure have long been examined to support the notion that participation is necessary and beneficial to human existence across the life course. A review of the literature reveals agreement among scholars that leisure does, in fact, provide measurable benefits in four broad areas of human life: individual and personal, social and community, environmental, and economic [1]. Alternatively, at the very least, leisure prevents a worsening condition from occurring in these areas [1–3]. Broadly speaking, leisure contributes to subjective well-being while also providing a host of physical, emotional, and psychological benefits for adults [1,4–8]. As defined by Sirgy [9], subjective well-being is a longer-term effective state that is comprised of three components: (a) actual experience of happiness or cumulative positive affect (joy, affection, pride, etc.) in salient life domains, (b) actual experience of depression or cumulative negative affect (sadness, anger, guilt, shame, anxiety, etc.) in salient life domains, and (c) evaluations of one's overall life or evaluations of salient life domains. This bottom-up model suggests that if one can improve overall leisure satisfaction, it will have a positive effect on subjective well-being [7]. This, in turn, can indirectly impact satisfaction in other domains, such as individual conditions, job satisfaction, and family satisfaction [2,6,8]. In addition, a myriad of physical, psychological, and emotional outcomes associated with these benefits can be realized [1,4,10–12].

During times of significant life change, leisure can also serve as an effective coping mechanism for major stressors by providing positive physiological, psycho-physiological,

psychological, and social benefits [13–17]. One such example can be seen in the changes associated with the developmental transitions into older adulthood, as individuals often experience substantial readjustments to their social identity structure and dramatic reductions in social convoys. In these instances, leisure has been used to successfully buffer the accompanying stressors later life can present [18,19]. For example, those traversing stages of grief following spousal death have used leisure to adapt to widowhood, manage elevated levels of stress, and find social support during their bereavement period [20–23]. The COVID-19 pandemic created widespread disruption to leisure participation, fractured social groups generated from leisure activities, and exacerbated risks to vulnerable groups, such as widows, with higher levels of need for social connectedness [19]. Considering leisure's impact in providing protective mechanisms during substantial change, the benefits of leisure social groups after spousal loss, and the pandemic-related disruptions to social group access emphasizes the importance of examining how effective leisure can be in preserving and maximizing well-being when normative life changes are complicated by non-normative events.

Understanding this process is especially relevant considering the profound impact of the recent pandemic on the global population. At the time of this article, there were more than 6.9 million deaths attributed to COVID-19 worldwide, including more than 1.1 million in the United States [24]. This staggering loss has resulted in an estimated 8.8 million people losing what they would consider to be a close family member [25]. Older adults are disproportionately impacted, particularly women, as data suggests a higher mortality rate in men, thus increasing the likelihood that a woman would have experienced spousal death [26,27]. For instance, approximately 34% of American women are navigating widowhood compared to 11.9% of American men [28]. Following a spousal loss, researchers found an elevated risk of adverse effects on the mental health of the surviving spouse, which could lead to worsened physical condition, including an increase in illnesses, greater use of medications, and poorer health ratings in general [25,26,29,30]. These negative effects are worsened among women, who have shown a greater reluctance to utilize help from others compared to male caregivers [31]. This reluctance to link with external sources of support further removes women from connections to others who not only provide support during caregiving but also during the subsequent bereavement period that can extend many years following spousal death. Adding an additional burden is the increased likelihood that spouses are sought to provide care for infirmed loved ones at the end of life. Spousal care for those with terminal illnesses is particularly isolating and challenging due to the highly complex nature of end-of-life caregiving.

### 1.1. Hierarchical-Compensatory Theory of Social Supports

Individuals who require elevated levels of support to complete daily tasks typically seek out informal support first, which consists of unpaid help from friends and family. Once their needs become greater than what can be provided by a person's informal support, they may then seek out formal support options, which consist of paid help from professionals. It is estimated that approximately 34.2 million Americans have been unpaid, informal caregivers to older adults [32]. The prevalence of informal caregiving has captured the attention of scholars over time and continues to be of high importance in contemporary societies. To better understand how families and individuals navigate systems of care, researchers have developed several models of caregiving processes. In general, individuals tend to seek out people who are closest to them in both emotional and physical proximity contexts. The hierarchical-compensatory theory of social supports [33] posits that a person in need of care typically considers a hierarchy of individuals available to provide such care and that family members are preferred over unrelated persons. If family members are not available or able to provide care, then a person will move outward to compensate for the inability to secure their first-choice caregivers. As pointed out by Cantor and Brennan [34], the hierarchy of individuals is constructed based on several factors, such as "psychological feelings of closeness, normative values, such as filial responsibility and family solidarity,

and the history of the relationship between the older person and other parties involved" (p. 36). Kinship care is preferred regardless of the circumstances [34], and spouses are sought first, followed by children [35]. The most recent set of data maintained by the Family Caregiver Alliance [32] outlines that "upwards of 75% of all caregivers are female and may spend as much as 50% more time providing care" (para 11). Additionally, those who care for a spouse are more likely to engage in high-hour caregiving, where they dedicate 21 or more hours per week. The COVID-19 pandemic created unique situations for families worldwide as it increased the need for caregiving while simultaneously reducing the surrounding resources for caregivers. "In many families, all caregiving responsibility was downsized to a smaller circle of caregivers within the family. Often, one family member assumed all responsibilities to reduce the risk of exposure and spread of COVID-19 [36]. The heightened risks for COVID-19-related mortality further complicated informal caregiving, considering the elevated needs surrounding terminal, chronic, and end-of-life care. Combining the literature on familial caregiving overtime with the unique circumstances of COVID-19 emphasizes the importance of considering the exact ways in which women caregivers experienced the care-providing process with dying spouses.

*1.2. Current Study*

The current study will explore the role of leisure in helping widowed caregivers negotiate the bereavement experience following spousal death in a post-pandemic society. These individuals are of particular interest as they are not yet represented in the extant literature, and they face elevated risks for social isolation because of aging, coupled with the responsibilities of spousal caregiving and exacerbated by the aggressive mitigation strategies necessitated by COVID-19. Specifically, we aim to understand if this highly vulnerable group, which is already at risk of social isolation due to typical developmental transitions (i.e., aging, retirement, decrease in social convoys, etc.), were able to experience the measurable benefits of leisure participation within the non-normative context of a global pandemic that directly limited or eliminated many leisure opportunities.

## 2. Methodology

*2.1. Participants*

To participate in the study, individuals had to meet specific participant criteria, which included being over the age of 60, identifying as women, and having experienced spousal loss during the COVID-19 pandemic. Both participants of this study met these requirements as both identified as women, were over the age of 60, and had experienced spousal loss before the declared end of the pandemic. Both women were also Caucasian and retired. Participant 1 ("Elizabeth") lives in a more densely populated suburban town outside of a small city in the Midwest of the United States. She is highly educated, holding both a bachelor's and master's degree, and is considered to be upper-middle socioeconomic status. Participant 2 ("Patricia") resides in a rural area with relatively low population density in the southeastern portion of the United States. She has a high school diploma and is considered to be of low-to-middle socioeconomic status. Participants were recruited for this study with the use of a non-probability sampling technique of snowballing.

*2.2. Data Collection*

A phenomenological multi-case study technique was used to explore the "multiple realities" [37] among women caregivers whose social convoys suffered pre-pandemic because of spousal caregiving and were further impacted by the COVID-19 mitigation strategies during the pandemic. Phenomenological multi-case study techniques have been used in the literature, e.g., [38–40], to garner a holistic understanding of relatively unknown groups while also allowing comparative elements to elucidate unique pathways within these small communities. Phenomenology celebrates the idea that different people undergoing similar circumstances will likely have different perceptions of their own lived experiences. In this way, phenomenology rejects the more traditional positivistic approach that expects

universal realities in favor of celebrating the diversified pathways to and through life experiences [37]. Data collection was informed by a transcendental phenomenological methodology [41]. As such, the interviews began by reiterating the purpose of the current study and asking participants to respond to one broad opening prompt: *In thinking about navigating COVID-19 and the loss of your spouse, what has your life been like?* The purpose of the opening prompt was to gather "data that will lead to a textural description and a structural description of the experiences, and ultimately provide an understanding of the common experiences of the participants" [42]. Subsequent questions throughout the interview were asked only when necessary and formulated directly from participant-offered content. Following each interview and the departure of the participants, the researchers engaged in a notation process to document any pertinent information not verbalized during the interview (e.g., body language, emotional cues, etc.) that may contribute to outlining the essence of participants' lived experiences. All procedures described in this section were approved following a review by the Institutional Review Board (IRB) of the researcher's academic institution (Protocol #: H23249). This review determined that the research subjects were at minimal risk, appropriate safeguards were planned, and the research activities involved only procedures that were allowable.

### 2.3. Analysis

Following data collection, a two-cycle approach of analysis was utilized, as advocated by Saldana [43], due to its efficiency in triangulating qualitative results. In the initial cycle, the researchers independently immersed themselves in the raw data gathered during participant interviews and engaged in open coding to discern concepts, patterns, themes, and categories associated with the multiple realities among the widows whose leisure participation was affected due to the strain of spousal caregiving and exacerbated by COVID-19 mitigation strategies. This initial coding process was exploratory and flexible in nature, thus allowing the researchers to create a general list of codes that captured the essence of the data. After completing the first cycle, the researchers then moved to the second phase, thematic analysis, which sought to develop a more structured framework by which to understand the data by identifying, analyzing, and reporting patterns (e.g., themes) within it. This collaborative process was designed to highlight disparities and divergent interpretations, which were then addressed through discussion and consensus-building by the researchers [44]. This two-cycle approach used the multiple viewpoints and expertise of the researchers to bolster the validity and reliability of qualitative findings, reducing individual bias and bolstering confidence in the results [43]. Following these steps, a qualitative comparison technique was employed to deepen our understanding of individuals' lived experiences [45–48] by elucidating potential differences in the multiple realities across participants. Qualitative comparison techniques have been deemed as adding a layer of rigor and complexity to research designs that seek to understand a holistic process from multiple realities [49].

### 3. Results

Thematic findings revealed three areas of similarities: Social isolation from caregiving, using leisure to self-focus and explore future possibilities, and core groups. Findings from the qualitative comparison are then offered to allow for a deeper understanding of the differences between participants' lived experiences. Pseudonyms will be used in place of participants' actual names to protect their anonymity. Participant one will be referred to as Elizabeth, and participant two will be referred to as Patricia.

### 3.1. Social Isolation from Caregiving

The participants reflected on being socially isolated as a result of spousal needs regardless of the restrictive mitigation strategies of the COVID-19 pandemic. Their ability to connect with others was limited as necessitated by their role as primary caregivers and the responsibilities thereof and not because of social distancing and lockdown requirements.

As participants noted, caregiving demands dominated what would happen next in their daily lives.

*"Before he passed away, I was home most of the time. I would go out to church, get groceries, and exercise, but only for short periods of time, because he was too sick to leave for very long...and doctor's appointments. [He] had to go to a lot of doctor's appointments, actually a whole lot of doctor's appointments, and that was kind of the social aspect [of my life]."*

~Elizabeth

*"COVID didn't bother us, we didn't go anywhere anyway. Our life didn't change much. I had to be there, but it didn't hinder my social life because we never had one before."*

~Patricia

*3.2. Using Leisure to Self-Focus and Explore Future Possibilities*

Following the death of their spouses, the participants reported an increased ability to self-focus and explore future possibilities while indicating the role that leisure would play. Specifically, leisure would be utilized to explore their newfound freedom and re-engage with themselves following the death of their spouses. Additionally, there were increased references to their futures and the ability to focus on themselves, a prospect that had not been afforded due to their role as primary caregivers.

*"Life has drastically changed. I go to more places, I socialize, I try new things! I'm always trying something new because now there's the availability to do that."*

~Elizabeth

*"It changed my whole world. I love it now, it's great, it's just wonderful. I want to take what's left of my life and move on. 48 years ago, was the last time I put myself first! Now, I can do the things that I want to do. I'm a free-range chicken!"*

~Patricia

*"One thing I'm super looking forward to is being able to walk right!"*

~Elizabeth

[Note: The participant reflected on knee replacement surgery needed years ago, but caregiving demands were prioritized. At the time of data collection, the participant had scheduled surgery for her knee to aid in her ability to walk correctly and pain-free.]

*"Life's just better. It feels glorious to put myself first."*

~Elizabeth

*"I focus on me because I can't help him anymore"*

~Patricia

*3.3. Core Groups*

Core groups were essential to both participants to help navigate day-to-day living during their role as primary caregivers and during the bereavement window to help with logistical and household responsibilities. In addition, both core groups played a vital role in satisfying emotional needs both during the caregiving and bereavement periods.

*"I was blessed with good friends and a lot of support, and people would come by and help, and so it made it a lot easier...I have a core group of 4 friends who are probably my base, who were always my base, and continue to be my base, and that has made and makes life a lot easier. We go to lunch, meet up, and do a lot of things together. Some friends I don't have anymore...I lost contact with them when he was sick."*

~Elizabeth

*"I have my kids. They helped me mow the yard and clean things out but I'm doing all that myself now. We can be open about how we feel, sometimes they are down and I'll*

*help them or sometimes I'll be down and they help me. I don't know how people make it without kids."*

~Patricia

It is expected that these core groups will remain intact and that new people are less likely to infiltrate the core group.

*"Those deep friendships take years to develop and I'm too old to socialize and make those happen."*

~Elizabeth

*"Interactions with neighbors and friends are different, I still like being by myself, I do better by myself, I want few people in my life."*

~Patricia

### *3.4. Differences*

The qualitative comparison technique generated some important differences surrounding core group composition and the ways in which leisure was experienced. Elizabeth's core group included four unrelated friends, and Patricia's core group were her two adult children. Core groups played slightly different roles and may contribute to the differences regarding the types of leisure activities enjoyed by each participant. Specifically, Elizabeth enjoyed more social leisure activities and expressed excitement to try new activities alongside her friends, as indicated by statements such as "*I socialize more with friends*", "*It's good to socialize with people at the Y [YMCA] which is why I go the times that I go. I even joined a club with older people!*", and finally, "*Now, it's my dream to go on an Alaskan cruise!*". Patricia preferred more individualistic pursuits, such as focusing on making her home and yard spaces more her own, which was represented in comments such as "*I tore down the barn, sold stuff, threw away stuff and just focused on getting things my way and making it my home.*" Patricia also reflected more of a filial obligation to her adult children to compensate for their loss of their father and this was likely driven by the shared grief experience wherein both parties were navigating the loss of a central family member. Specifically, Patricia mentioned, "*I try to be as strong as I can for them because they still have me. I'm trying to be as strong as I can. He was the strong one. I have to be a little bigger for them to look up to. I need to protect them*".

## 4. Discussion

End-of-life caregiving and spousal death require a multitude of difficult adjustments, even in situations where death is expected and among couples with imperfect relationships [50]. There is often a marked shift away from established spousal roles to those of a primary caregiver. This new role is particularly challenging and isolating due to the highly complex nature of end-of-life care and can become an all-encompassing activity that dominates one's life [51]. After a sustained period of illness and caregiving, family members often feel a mixture of relief and grief once their loved one has passed. Relief from the strain of caregiving and end to the suffering endured by their loved ones, but also the grief of a family member lost, especially for a primary caregiver [51]. Following a spousal loss in particular, the researchers found a long-term, elevated risk of adverse effects on the mental health of the surviving spouse, which could lead to worsened physical condition, including an increase in illnesses, greater use of medications, and poorer health ratings in general [32,35,36]. These negative effects are worsened among women, who have shown a greater reluctance to utilize help from others compared to male caregivers [37]. Therefore, the women in this study are not only negotiating widowhood, which is considered to be a major developmental transition due to the many life changes it brings, but are doing so after a prolonged period of social isolation further complicated by the non-normative event of the COVID-19 pandemic.

Findings from the current study revealed that both participants experienced isolation from spousal caregiving, utilized leisure to negotiate the subsequent developmental transition to widowhood, and relied on core groups to help navigate day-to-day living

during their role as primary caregivers and during the bereavement window. However, the expectation that isolation would be further exacerbated by the COVID-19 pandemic and subsequent mitigation strategies and that leisure would be used to cope with spousal death did not materialize. Both participants indicated that the effects of mandated lockdowns, social distancing, and other protocols were minimized in their lives and leisure since they were already removed from general society due to caregiving responsibilities. This speaks to the magnitude of spousal caregiving demands and the constraints on leisure participation, wherein a pandemic did not significantly modify daily life for these women. Put another way, caregiving already had a pandemic-like effect before the pandemic ever surfaced. As for leisure and its role in coping, research tells us that people use leisure in many beneficial ways during major life changes, and it was expected that participants would rely on leisure in these traditional ways. It was, therefore, a surprise to learn that leisure played a minimal role in coping and instead was more transformative and transitional.

A qualitative comparison discovered some important differences in the surrounding core group composition and the ways in which leisure was experienced during this time. For the first participant, Elizabeth, the core group was composed primarily of four unrelated friends. For the second participant, Patricia, the core group was made up of her two adult children. While the functions of the groups were similar, the dynamics and perceived responsibilities of the participants toward their groups were different. For Elizabeth, this was primarily a friend group and was treated as such. For Patricia, there was an added responsibility of a parent who is not only relying on her children to act as a support structure but also feels an obligation to support them as they negotiate a transition of their own (i.e., the death of their father). We postulate that the group construction and dynamics are heavily influenced by the environment in which they live. Elizabeth resides in a suburban environment with her core group living in relative proximity; thus, she can establish, build, and engage in these relationships more easily. Patricia, in contrast, lives in a decidedly rural location where any potential non-family friend group is spread out and not readily accessible. It is not uncommon for older adults living in this type of environment to be more isolated and have difficulty constructing a non-family social network [52–55]. This also could explain why Elizabeth was able to rely more heavily on social leisure while Patricia was more individualistic in her activities.

Even though the many stressors of widowhood and risks for negative effects on a person's well-being are well documented see [34], the participants in the current study expressed some positive ways in which their lives had transformed. Prior to the death of their spouses, the participants were unable to envision much beyond the walls of the relationship and daily caregiving demands. However, several months into their bereavement periods, both participants described their lives as a time of self-focus and budding freedom. Specifically, this time was filled with contemplation about the many directions their futures could go and a newfound sense of autonomy and control they now have over their lives that was not present during the caregiving-intensive period. From a life course development framework, much is to be said about these findings within the context of human development. Particularly, the theme of using leisure to self-focus and explore future possibilities aligned with aspects of the developmental period of emerging adulthood [56]. Emerging adulthood demarcates a time in life filled with major change and developmental transition. Typically occurring during a person's 20s, this time in life is accompanied by major exploration in the realms of self, love, work, and worldviews. People experience a type of in-between feeling where they are making independent, adult-like decisions in some realms of life but still dependent on their families for certain resources. Arnett [56] pinpointed the following five defining dimensions of emerging adulthood: feeling in between an adolescent and adult status, age of identity exploration, time for instability, time of self-focus, and age of possibilities. Interestingly, participants in the current study described their lives in ways that mapped to the emerging adulthood dimensions of time of self-focus and age of possibilities. The self-focused nature of this developmental period is afforded to individuals because of the lack of daily commitments to others and involves

both large-scale (e.g., career-related decisions, etc.) and small-scale (e.g., whether to come home at night, etc.) decisions. Participants in the current study were freed from a highly demanding schedule ingrained within their relationships and reveled in the expansion to self-focus decisions in both large and small-scale choices. They referenced large-scale thoughts, such as contemplating where they may live in the future, among other big decisions, and they also discussed many smaller choices, such as where, when, and what to eat. In fact, one participant even commented in awe that she could go out for food at a 24 h restaurant if she were unable to sleep at night, and this choice had never been an option before. The widowed caregivers in the current study were embarking on a type of soul-searching journey wherein they self-focused to better understand their needs and make choices to best suit what they hoped to gain out of life. As captured in the emerging adulthood framework [56,57], this manifested in both small and large ways.

Another shared similarity between the emerging adulthood period and the experiences of participants was a sense that the current time in life was an age of possibilities. This is an opportunity for transformation, robust decision-making, and excitement about the future. Arnett [57] even discusses that the age of possibilities is increasingly meaningful for individuals who are exiting complicated family situations as it presents a new freedom for decision-making that could significantly enhance their futures. These descriptions were heavily represented in the lives of the participants in the current study. Both participants reflected on the dramatic transformations their lives had already endured and contemplated future changes with excitement. The nature of the spousal relationships for both participants was not conducive to envisioning a realm of future possibilities. Instead, these women were dominated by the more immediate demands of their daily lives and spousal caregiving. Now, for the first time in decades, they have a new freedom to focus on their own futures.

Widowhood is undoubtedly associated with many negative outcomes, but participants in the current study also reported more positive aspects surrounding liberation, personal growth, and excitement for the future. Positive experiences submerged within catastrophic change have surfaced in the literature [58] but remain a relatively understudied phenomenon. Extant literature would benefit from a greater understanding of these experiences in order to extend our knowledge of diversified bereavement pathways. Additionally, special attention to the needs of caregivers within a social well-being context is warranted. The majority of the literature focuses broadly on the negative effects of social isolation among the general population of older adults. The current study yielded results for a specific subgroup where the act of caregiving created pandemic-like isolation even before the onset of the actual pandemic. Current age and health trends across the world suggest that caregiving will become more common in the future and necessitates a better understanding of how the many benefits of leisure can be actualized by specific subgroups, especially considering the increased importance of leisure participation as people age [7]. This is especially important to adult women who may have elevated needs for social support during their bereavement window following a period of intensive caregiving and high relationship demands.

## 5. Conclusions

Social isolation in later life is typically focused on from the perspective of the care recipient, while less attention has been given to the social isolation effects from the perspective of the caregiver. The herculean tasks of caregiving are well documented see [58–61], but the current study highlighted the magnitude of relatively normative tasks within non-normative global contexts. Moreover, this study continues to expand our understanding of the role of leisure across the lifespan. To date, there has been little investigation on the ways in which leisure provides benefits during later life, much less among highly vulnerable populations during the bereavement window following spousal loss. The current study shed light on the multiple realities of women caregivers following the death of their spouses during unprecedented times and explored the ways in which leisure impacted their grief

journeys. It was discovered that the return to normalcy following the pandemic would have been irrelevant had the caregiving tasks not been lifted. This speaks to the true magnitude of caregiving responsibilities, which acted as significant barriers to social connections. In this way, being liberated from the constraints of caregiving allowed participants to use leisure as avenues to self-focus and explore future possibilities. Although focusing on such outcomes following death violates social expectations placed on the bereaved, these findings add to a small body of growing research, see [62,63] that highlights this often undiscussed positive phenomenon.

Limitations of the current study revolve around the small size of the sample. Although small sample sizes of two participants are appropriate for phenomenological methodologies and are appropriate in instances where the target population is a highly unique set of individuals, small sample sizes hinder the generalizability of findings. Now that a foundational framework for understanding the relevance of leisure in the lives of socially isolated women caregivers navigating widowhood in a post-pandemic society has been laid, future research would benefit from expanding this line of inquiry within a broader range of participants. Understanding the ways in which leisure manifests during bereavement periods for highly isolated individuals will help scholars better understand the full impact that leisure can offer to grieving persons and how leisure presents itself within diverse groups.

**Author Contributions:** Conceptualization, T.P.S. and J.Z.; methodology, T.P.S. and J.Z..; validation, T.P.S. and J.Z.; formal analysis, T.P.S. and J.Z.; investigation, T.P.S. and J.Z.; resources, T.P.S. and J.Z.; data curation, T.P.S. and J.Z.; writing—original draft preparation, T.P.S. and J.Z.; writing—review and editing, T.P.S. and J.Z.; visualization, J.Z. All authors have read and agreed to the published version of the manuscript.

**Funding:** This research received no external funding.

**Institutional Review Board Statement:** The study was conducted in accordance with the Declaration of Helsinki and approved by the Institutional Review Board of Georgia Southern University (protocol code H23249, approved on 24 March 2023).

**Informed Consent Statement:** Informed consent was obtained from all subjects involved in the study.

**Data Availability Statement:** The data are not publicly available to protect the privacy and confidentiality of study participants.

**Conflicts of Interest:** The authors declare no conflict of interest.

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
