# Peer review of "Examining the Role of Leisure in Navigating Spousal Death: A Phenomenological Multi-Case Study of Widowhood"

_2673-9259, doi:10.3390/jal3040021_

Round 1

Reviewer 1 Report

Comments and Suggestions for Authors

Thank you for nominating me as a reviewer. In this paper authors explore the post-pandemic bereavement window of widowed care-givers who experienced higher degrees of social isolation and the ways in which leisure was used to cope. In this paper, a phenomenological multi-case study technique was used to explore the “multiple realities'' among female caregivers whose social convoys suffered pre-pandemic because of spousal caregiving and were further impacted by COVID-19 mitigation strategies during the pandemic. This paper is overall well written. If minor revisions are completed by the authors, the quality of the study will be further improved.

1. Do you need subheadings in the introduction section? I leave it to the authors' judgment.

2. The authors need to describe the procedures for the phenomenological multiple case study method in more detail.

3.It would be a good idea to add a conclusion section.

Author Response

  1. The addition of subheadings in the introduction

Author response: The formatting of the paper was examined throughout and updated to be more organized (see pg. 4).

  1. The authors need to describe the procedures for the phenomenological multiple case study method in more detail.

Author response: The methodology section was extensively revised to explain the phenomenological multi-case study method more adequately in more detail beginning on page 4, line 148.

  1. It would be a good idea to add a conclusion section.

Author response: A conclusion paragraph was added which not only synthesized key points of the study, but also detailed limitations and future research related to this topic (see pg. 8, line 405). 

Reviewer 2 Report

Comments and Suggestions for Authors

Dear authors.

I think that the article presented is interesting in terms of the subject matter; however, I believe that some aspects need to be improved.

The introduction presents the subject well and the theoretical foundation is adequate.  It would be interesting to explain more clearly and precisely the objective of the study, as well as the sample (number, age, etc.).  The instruments used should be presented and explained more clearly (questions that make up the interviews, etc.).  It would have been appropriate to use a mixed design or, in any case, to extend the qualitative instruments with focus groups.

Careful with the inferences.  Situations are being described.

The conclusions and limitations of the research are missing.

Best regards.

Author Response

Reviewer 2

  1. The introduction presents the subject well and the theoretical foundation is adequate.  It would be interesting to explain more clearly and precisely the objective of the study.

Author response: The objective study was more clearly laid out in a revised ‘Current Study’ section on page 3.

  1. The instruments used should be presented and explained more clearly (questions that make up the interviews, etc.)

Author response: The nature of a phenomenological methodology are conducive to only one general opening prompt which was provided in the methods section of the paper (pg. 4, line 148)

  1. It would have been appropriate to use a mixed design or, in any case, to extend the qualitative instruments with focus groups.

Author response: This is a valuable suggestion but was not possible given the exploratory nature of the current study in utilizing a phenomenological technique to explore the circumstances within unknown populations. We have represented this suggestion in the future research portion of the paper (see pg. 9).

Reviewer 3 Report

Comments and Suggestions for Authors

Justification of the Methodology: Perhaps a more detailed and justifiable explanation was given as to why the phenomenological method and the multi-case study were specifically chosen, and how these methods allow us to explore the "multiple realities" mentioned.

Consistency in the editorial: There is a linguistic inconsistency that could distract or confuse readers, for example, "who's" should be "whose".

Museum Details: More details could be given on how cases were selected for the multi-case study, because these cases are particularly informative or relevant to investigation questions, and any other consideration of the museum.

More Details in Data Analysis: Even if we mention the use of a focus of two cycles and the codification of patrons, it would be useful to provide more details about how this process was carried out, thus providing specific examples of the analysis in action.

Verification of Data Validation: It is essential to discuss how investigators validated data. For example, do you use triangulation or member checking to ensure the validity of the data and results?

Description of Participants: We may include a more detailed description of the participants so that readers can better understand the people involved.

Exploration of Subjects and Limitations: Although this may be discussed in another section of the paper, it is essential to be transparent regarding the possible aspects of the investigator and limitations of the study.

Connection between Data and Themes: You may find examples of quotes or direct interactions of the connection of data that illustrate how the identified themes were derived or justified.

Clarity in Presentation: Make sure that the descriptions of the methods and the hallmarks are clear and accessible to a wide audience of readers.

However, it is best to explain the context, such as religion, ethnic group, location of the study, cultural or educational level of the participants of the study, without highlighting the conclusions or limits of the study, in order to be accepted, profound changes must be made in the research method

Author Response

  1. Comments and Suggestions for Authors Justification of the Methodology: Perhaps a more detailed and justifiable explanation was given as to why the phenomenological method and the multi-case study were specifically chosen, and how these methods allow us to explore the "multiple realities" mentioned.

    Author response: The analysis plan and data collection sections have been reworked to better articulate these methodological portions of the study (see pg. 4). 

  1. Consistency in the editorial: There is a linguistic inconsistency that could distract or confuse readers, for example, "who's" should be "whose."

Author response: The paper was reexamined, and efforts were made to correct any issues and eliminate inconsistencies.

  1. More Details in Data Analysis: Even if we mention the use of a focus of two cycles and the codification of patrons, it would be useful to provide more details about how this process was carried out, thus providing specific examples of the analysis in action. & Verification of Data Validation: It is essential to discuss how investigators validated data. For example, do you use triangulation or member checking to ensure the validity of the data and results?

Author response: The Analysis section has been expanded to elucidate the needs outlined by Reviewer 3 (see page 4). 

  1. Description of Participants: We may include a more detailed description of the participants so that readers can better understand the people involved.

Author response: A detailed description of participants was added to a heavily revised methodology section beginning on pg. 3. 

  1. Exploration of Subjects and Limitations: Although this may be discussed in another section of the paper, it is essential to be transparent regarding the possible aspects of the investigator and limitations of the study.

Author response: Efforts to address limits has been added to the conclusion section of the manuscript (see pgs. 8-9).

  1. Connection between Data and Themes: You may find examples of quotes or direct interactions of the connection of data that illustrate how the identified themes were derived or justified.

Author response: Direct quotes and excerpts were provided to represent the participant-specific contributions that were analyzed and subsequently led thematic development.

  1. However, it is best to explain the context, such as religion, ethnic group, location of the study, cultural or educational level of the participants of the study, without highlighting the conclusions or limits of the study, in order to be accepted, profound changes must be made in the research method.

Author response: There have been extensive changes made to multiple areas of the manuscript to enhance clarity and ensure that adequate detail is available to the readership, so that they may understand the full realm of this work. In doing so, we have addressed these areas of concern presented by Reviewer 3.

Round 2

Reviewer 2 Report

Comments and Suggestions for Authors

Dear Authors.  I believe that an effort has been made to respond to the recommendations of the reviewers, adjusting the final result to the minimum requirements for a scientific publication.  Although the limitations of the study are clear, I believe it can be published.

Kind regards.

Author Response

My co-author and I sincerely appreciate the opportunity to make final enhancements on our manuscript as we look forward to moving forward with publication in your Journal. We have taken considerable attention to the final version to make the recommended change to line 396 as well as minor updates in other areas. We believe we have exhausted the need for final revisions and look forward to continued communication with you to see our paper fully through the process.
